# Occurrence and Genomic Characterization of Clone ST1193 Clonotype 14-64 in Uncomplicated Urinary Tract Infections Caused by *Escherichia coli* in Spain

Isidro García-Meniño,[a,b,c] Pilar Lumbreras,[d,e] Luz Lestón,[a,b] Mónica Álvarez-Álvarez,[d] Vanesa García,[a,b] ⓘ Jens Andre Hammerl,[c] ⓘ Javier Fernández,[d,e,f,g] ⓘ Azucena Mora[a,b]

aLaboratorio de Referencia de Escherichia coli (LREC), Dpto. de Microbioloxía e Parasitoloxía, Universidade de Santiago de Compostela (USC), Lugo, Spain
bInstituto de Investigación Sanitaria de Santiago de Compostela (IDIS), Santiago, Spain
cDepartment for Biological Safety, German Federal Institute for Risk Assessment, Berlin, Germany
dServicio de Microbiología, Hospital Universitario Central de Asturias (HUCA), Oviedo, Spain
eGrupo de Microbiología Traslacional, Instituto de Investigación Sanitaria del Principado de Asturias (ISPA), Oviedo, Spain
fResearch & Innovation, Artificial Intelligence and Statistical Department, Pragmatech AI Solutions
gCIBER de Enfermedades Respiratorias (CIBERES), Instituto de Salud Carlos III, Madrid, Spain

Javier Fernández and Azucena Mora contributed equally to this work. Author order was determined based on seniority.

**ABSTRACT** We conducted a prospective, multicenter, specific pilot study on uncomplicated urinary tract infections (uUTI). One-hundred non-duplicated uropathogenic *Escherichia coli* (UPEC) from uUTI occurred in 2020 in women attending 15 primary care centers of a single health region of northern Spain were characterized using a clonal diagnosis approach. Among the high genetic diversity showed by 59 different phylogroup-clonotype combinations, 11 clones accounted for 46% of the isolates: B2-ST73 (CH24-30); B2-ST73 (CH24-103); B2-ST131 (CH40-30); B2-ST141 (CH52-5); B2-ST372 (CH103-9); B2-ST404 (CH14-27); B2-ST404 (CH14-807); B2-ST1193 (CH14-64); D-ST69 (CH35-27); D-ST349 (CH36-54), and F-ST59 (CH32-41). The screening of the UPEC status found that 69% of isolates carried ≥ 3 of *chuA, fyuA, vat*, and *yfcV* genes. Multidrug resistance to at least one antibiotic of ≥ 3 antimicrobial categories were exhibited by 30% of the isolates, with the highest rates of resistance against ampicillin/amoxicillin (48%), trimethoprim (35%), norfloxacin (28%), amoxicillin-clavulanic acid (26%), and trimethoprim-sulfamethoxazole (24%). None extended-spectrum beta-lactamase/carbapenemase producer was recovered. According to our results, fosfomycin and nitrofurantoin should be considered as empirical treatment of choice for uUTI by *E. coli* (resistance rates 4% and 2%, respectively). We uncover the high prevalence of the pandemic fluoroquinolone-resistant ST1193 clone (6%) in uUTI, which represents the first report in Spain in this pathology. The genomic analysis showed similar key traits than those ST1193 clones disseminated worldwide. Through the SNP comparison based on the core genome, the Spanish ST1193 clustered with isolates retrieved from the Enterobase, showing high genomic similarity than the global ST1193 described in the United States, Canada and Australia.

**IMPORTANCE** Analyzing the clonal structure and antimicrobial resistance of *E. coli* isolates implicated in uncomplicated urinary tract infections, one of the most frequent visits managed in primary health care, is of interest for clinicians to detect changes in the dynamics of emerging uropathogenic clones associated with the spread of fluoroquinolone resistance. It can also provide consensus concerning optimal control and antibiotic prescribing.

**KEYWORDS** uncomplicated UTI (uUTI), antimicrobial resistance (AMR), UPEC, ST1193, ST131, fluoroquinolone resistance (FQR), *Escherichia coli*

Address correspondence to Azucena Mora, azucena.mora@usc.es.

The authors declare no conflict of interest.

[This article was published on 23 May 2022 with an error in the Discussion section. The Discussion section text has been updated in the current version, posted on 26 May 2022.]

Urinary tract infections (UTIs) are one of the most common diseases in both community and hospital settings. In the United States, UTIs were estimated in approximately 400,000 hospitalizations and $2.8 billion of costs in 2011 (1). In France, it was estimated in a mean of 70 € for one episode of a suspected UTI and 58 million from the societal perspective (2). In particular, uncomplicated urinary tract infections (uUTIs) are within the most frequent visits managed in primary health care, which implies a high impact on antibiotic prescription in ambulatory medicine (15% to 20%) (3–6). uUTI are referred as episodes of acute cystitis occurring in healthy, premenopausal, and nonpregnant women with no known structural or functional urological abnormalities of the genitourinary tract. The most frequent symptoms associated with uUTIs are strong and persistent urge to urinate, pelvic pain, burning sensation when urinating, and sign of blood in the urine (7). In Spain, a prospective study at two primary care practices in 2007 found that infectious diseases accounted for one third of the visits to a general practitioner. Among 4,353 infections observed in patients with a mean age of 44.1 years, UTIs were the third cause (452; 10.4%), after respiratory (2,196; 50.4%) and skin infections (586, 13.5%) (8).

*Escherichia coli* is the most commonly etiological agent reported in UTIs (75% to 95%) (9). Unfortunately, there is a dramatic increase of multidrug resistance (MDR) within uropathogenic *E. coli* (UPEC), mainly due to a few globally disseminated clones such as ST131-*H30R* or ST1193-*H64*, which further compromises treatment (7, 10–16). While isolates causing complicated UTI have been widely analyzed and reported (17–19), studies on uUTIs are still lacking because urine culture is rarely included in the routine diagnosis. In fact, at clinical level, the European Association of Urology and the Infectious Diseases Society of America (IDSA) recommends the treatment for uUTIs based on an empirical antimicrobial therapy guided by local susceptibility patterns without urine culture (20). Therefore, microbiological urine testing prior to treatment is only essential for patients with complicated UTIs (structural or functional abnormalities in genitourinary tracts), or when empirical therapy fails (10). In consequence, microbiological data lack for up to 70% of females with UTI symptoms. Based on a four-country primary care cohort study, Butler et al. (21) concluded that variation in management of uUTI at a country primary care network level is clinically unwarranted and highlights a lack of consensus concerning optimal symptom control and antibiotic prescribing.

Based on what was stated above, we aimed the characterization of the predominant uropathogenic *E. coli* (UPEC) lineages associated with uUTIs to assess the importance of implementing specific surveillance programs to treat these infections.

## RESULTS

**Virulence status.** The screening of virulence traits associated with a higher efficiency in the colonization of the urinary tract revealed that 69 of the 100 uUTI isolates conformed the UPEC status ($\geq$ 3 of specific virulence traits *chuA*, *fyuA*, *vat*, and *yfcV*). Individually, *chuA*, *fyuA*, *vat*, and *yfcV* were present in 91%, 91%, 59%, and 67% of the isolates, respectively. Besides, the screening of *iutA* and *kpsMII* as prevailing genes within extraintestinal infections showed that 44% of the *E. coli* were carriers of both. Individually, they were detected in 55% and 80% of the isolates, respectively (Table S1).

**uUTI clonal groups.** The eight *E. coli* phylogroups *sensu stricto* (A, B1, B2, C, D, E, F, and G) were represented among the 100 uUTI isolates. However, B2 and D were the most prevalent, accounting for 67% and 14% of the isolates, respectively. The remaining 19 uUTI *E. coli* belonged to A (5%), B1 (5%), E (3%), F (3%), C (2%), and G (1%). The predictor genes associated with a higher efficiency in the colonization of the urinary tract, and therefore positive for the UPEC status was found within the isolates belonging to phylogroups B2 (65 of 67 isolates), F (3), and G (1). Both *iutA* and *kpsMII* virulence traits were present in 44 isolates of phylogroups A (2 out of 5 isolates), B2 (32 of 67), D (7 of 14), and F (3) (Table S1).

By clonotyping, 59 different phylogroup-clonotype (PG-CH) combinations were revealed, and those represented by $\geq$ 3 isolates were further investigated by multilocus sequence typing (MLST). As a result, 46% of the 100 uUTI isolates belonged to one

**TABLE 1** Prevalent clones (≥ 3 isolates) within the 100 uUTI *E. coli*, virulence traits, MDR, and FQR

| Phylogroup-ST | Clonotype (CH) | No. isolates (N = 46) | UPEC status (N = 36) | *iutA + kpsMII* (N = 26) | MDR (N = 16) | FQR (N = 10) |
|---|---|---|---|---|---|---|
| B2-ST141 | CH52-5 | 6 | 6 | 0 | 0 | 0 |
| B2-ST1193 | CH14-64 | 6 | 6 | 6 | 4 | 6 |
| B2-ST73 | CH24-103 | 5 | 5 | 0 | 1 | 0 |
| | CH24-30 | 3 | 3 | 3 | 2 | 0 |
| B2-ST372 | CH103-9 | 4 | 4 | 0 | 2 | 1 |
| B2-ST404 | CH14-27 | 4 | 4 | 4 | 0 | 0 |
| | CH14-807 | 3 | 2 | 2 | 0 | 0 |
| B2-ST131[a] | CH40-30 | 3 | 3 | 3 | 3 | 0 |
| D-ST69 | CH35-27 | 5 | 0 | 3 | 1 | 0 |
| D- ST349 | CH36-54 | 4 | 0 | 2 | 3 | 2 |
| F-ST59 | CH32-41 | 3 | 3 | 3 | 0 | 1 |

[a]Isolates belonging to the clonal group ST131 appeared associated to three clonotypes: CH40-30, CH40-22, CH40-1196 (with three, one, and one isolates, respectively). While only the three isolates of clone B2-ST131 (CH40-30) exhibited MDR, B2-ST131 (CH40-1193) showed FQR.

of the following 11 clones: B2-ST141 (CH52-5), six isolates; B2-ST1193 (CH14-64), six isolates; B2-ST73 (CH24-103), six isolates; B2-ST73 (CH24-30), three isolates; B2-ST372 (CH103-9), four isolates; B2-ST404 (CH14-27), four isolates; B2-ST404 (CH14-807), three isolates; B2-ST131 (CH40-30), three isolates; D-ST69 (CH35-27), five isolates; D-ST349 (CH36-54), four isolates; and F-ST59 (CH32-41), three isolates. Most isolates (36 of 46) belonging to nine of the 11 prevalent clones were carriers of ≥ 3 of specific traits *chuA*, *fyuA*, *vat*, and *yfcV* associated with higher virulence, and therefore, positive for the UPEC status; and only clones D-ST349 (CH-54) and D-ST69 (CH35-27) were characteristically negative. Regarding the presence or absence of both *iutA* and *kpsMII*, we observed a positive correlation with isolates belonging to clones B2-ST73 (CH24-30), B2-ST131 (CH40-30), B2-ST404 (CH14-27), B2-ST1193 (CH14-64), and F-ST59 (CH32-41). In addition to clonotype CH40-30, determined in three isolates, the pandemic ST131 clonal group appeared also associated to CH40-22 and CH40-1196 (one isolate each) (Table 1; Table S1).

**Antimicrobial susceptibility testing.** Antimicrobial susceptibility testing (AST) showed the highest rates of resistance to ampicillin/amoxicillin (48%), ticarcillin (45%), piperacillin (41%), trimethoprim (35%), norfloxacin (28%), amoxicillin-clavulanic acid (26%), and trimethoprim-sulfamethoxazol (24%). Furthermore, 30% of the uUTI isolates were MDR, being *in vitro* resistant to at least one antibiotic of ≥ 3 antimicrobial categories (Table 2). It is important to note that none of the 100 uUTI isolates exhibited phenotypic expression of ESBL enzymes. Regarding colistin, two isolates were categorized as resistant by the MicroScan System but as susceptible by the standard BMD method (both isolates displayed MICs of 0.125 mg/L).

The group of uUTI isolates positive for the UPEC status exhibited lower rates of MDR (18 of 69, 26%) but without significant differences in comparison to those negative for the UPEC status (13 of 31, 42%) ($P$ = 0.16). Particularly among the 11 prevalent uUTI clones, the global rate of MDR was 34.8% (16 out of 46 isolates): namely, B2-ST131 (CH40-30) (three isolates), B2-ST1193 (CH14-64) (four of six isolates), D- ST349 (CH36-54) (three of four isolates), B2-ST73 (CH24-30) (two of three isolates), B2-ST372 (CH103-9) (two of four isolates), B2-ST73 (CH24-103) (one of five isolates), and D-ST69 (CH35-27) (one of five isolates) (Table 1).

Fluoroquinolone-resistance (FQR) was exhibited by 28% of the 100 uUTI isolates belonging to 20 different PG-CH combinations, including the six isolates of clone B2-ST1193 (CH14-64), which were further analyzed by whole genome sequencing (WGS) due to the global emergence of this FQR and virulent lineage (Table 1; Table S1).

***In silico* characterization of B2-ST1193 (CH14-64) isolates.** Table 3 summarizes the main traits of the *in silico* characterization. All B2-ST1193 (CH14-64) isolates were subjected to WGS. However, one isolate that had less than 20 times coverage was excluded. SerotypeFinder predicted O75:H5 for all genomes but one (LREC-270), for

**TABLE 2** Rates of antimicrobial resistance interpreted according to EUCAST 2021 breakpoints

| Antimicrobial categories | Antimicrobial agent | % Resistant isolates N = 100 |
|---|---|---|
| Penicillins | Ampicillin/amoxicillin | 48 |
| | Piperacillin | 41 |
| | Ticarcillin | 45 |
| Antipseudomonal penicillins + beta-lactamase inhibitors | Piperacillin-tazobactam | 5 |
| Penicillins + beta-lactamase inhibitors | Amoxicillin/clavulanic acid | 26 |
| Non-broad spectrum cephalosporins: 1st & 2nd generation | Cefuroxime | 3 |
| Broad-spectrum cephalosporins: 3rd & 4th generation | Cefixime | 1 |
| | Cefotaxime | 0 |
| | Ceftazidime | 0 |
| | Cefepime | 0 |
| Broad-spectrum cephalosporins+beta-lactamase inhibitor | Ceftolozane-tazobactam | 0 |
| | Ceftazidime-avibactam | 0 |
| Carbapenems | Imipenem | 0 |
| | Ertapenem | 0 |
| | Meropenem | 0 |
| Monobactams | Aztreonam | 1 |
| Fluroquinolones | Norfloxacin | 28 |
| | Ciprofloxacin | 18 |
| | Levofloxacin | 17 |
| Aminoglycosides | Gentamicin | 17 |
| | Tobramycin | 16 |
| | Amikacin | 4 |
| Glycylcyclines | Tigecycline | 0 |
| Nitrofuran derivatives | Nitrofurantoin | 2 |
| Phosphonic acids | Fosfomycin | 4 |
| Folate pathway inhibitors | Trimethoprim | 35 |
| | Trimethoprim-sulfamethoxazole | 24 |
| Polymyxins | Colistin | 0 |
| MDR (resistant to ≥ one agent in ≥ 3 antimicrobial categories) | | 30 |

which O antigen was not typeable (NT:H5). Besides, MLST (seven genes) and CHTyper tools confirmed ST1193 (CH14-64) for all genomes. The alternative MLST scheme based on the eight genes *dinB*, *icdA*, *pabB*, *polB*, *putP*, *trpA*, *trpB*, and *uidA* (22) predicted ST53 (1-7-1-9-20-20-1-6) for all genomes but LREC-270 (ST53-like; *trpA*, 92% identity with allele 20). cgMLSTFinder also differenciated LREC-270, which was assigned as cgST7142, while LREC-265, 269, 273, and 275 were predicted as cgST4085.

All ST1193 genomes presented the same set of chromosomal mutations conferring FQR (*gyrA* S83L, *gyrA* D87N, *parC* S80I, and *parE* L416F), which correlated with the *in vitro* expression. Additionally, other acquired resistance traits were predicted using CGE and CARD databases (Table 3, Table S2). Using VirulenceFinder and VFDB, all genomes but LREC-270 showed the same profile, including a wide number of extraintestinal virulence genes (Table 3, Table S3). Likewise, MobileElementFinder determined the same profile of mobile genetic elements in correlation with AMR and virulence traits except for LREC-270. The plasmidome analysis revealed the presence of an IncF [F-:A1:B10] plasmid type in four of five genomes (all but LREC-270), together with small non-conjugative Col-like plasmids in four of them (Table 3). The five ST1193 genomes were predicted as human pathogen (probability > 92%) when analyzed with PathogenFinder. Finally, CRISPRCasFinder found the same types/subtypes of CRISPR-Cas system I within LREC-269, 270, and 275 (Table 3).

Enterobase (https://enterobase.warwick.ac.uk/) was searched as a source of genomes belonging to ST1193 (according to the Achtman 7-gene MLST scheme), predicted as cgST4085 and cgST72142, whose assemblies were retrieved for comparative

**TABLE 3** *In silico* characterization and phenotypic AMR of the uUTI ST1193 isolates

| ID code[a] | O:H antigens[b] | ST#1/ST#2[c] | cgST[d] | CHType[e] | Acquired resistances and point mutations (in bold)[f] | Plasmid content Inc. group [pMLST][g] | Virulence genes[h] | Mobile genetic elements (& relation to AMR and virulence traits)[i] | CRISPR-Cas system / Cas-type/subtype[j] | Phenotypic AMR[k] |
|---|---|---|---|---|---|---|---|---|---|---|
| 51107635 LREC-265 | O75:H5 | 1193/53 | 4085 | 14-64 | bla_TEM-1B; aph(3'')-Ib, aph(6)-Id; mdf(A), sul2; sitABCD **gyrA S83L, gyrA D87N, parC S80I, parE L416F** | IncF [F:-A1:B10] IncI1-I [ST Unknown] IncQ1* ColBS512-like | chuA, fyuA, gad, iha, irp2, iucC, iutA, kpsE, kpsMII_K1, neuC, ompT, papA_F43, sat, senB, sitA, terC, usp, vat, yfcV | ISEc42 (mdf(A), ompT); MITEEc1 (mdf(A), ompT); IS30 (sitABC, sitA); ISEc31 (iha); IS629 (papA_F43); MITEEc1 (fyaA, irp2, yfcV); MITEEc1 (terC) | - | AMP-AM; PIP; TIC; AMC; CIP; LEV; NOR |
| 51140663 LREC-269 | O75:H5 | 1193/53 | 4085 | 14-64 | bla_TEM-1B; aph(3'')-Ib, aph(6)-Id; mdf(A), mph(A), sul2, dfrA17; sitABCD **gyrA S83L, gyrA D87N, parC S80I, parE L416F** | IncF [F:-A1:B10] IncQ1* ColBS512-like ColRNAI-like* | chuA, fyuA, gad, iha, irp2, iucC, iutA, kpsE, kpsMII_K1, neuC, ompT, papA_F43, sat, senB, sitA, terC, usp, vat, yfcV | ISEc42 (mdf(A)); MITEEc1 (mdf(A)); MITEEc1 (terC); IS30 (sitABCD, sitA); MITEEc1 (irp2, yfcV, fyaA); ISEc31 (iha); IS629 (papA F43); IS100 (iucC, sat, iutA) | Class I / CAS-Type I/Cas3-0-I; CAS-Type IIIA/Csm2-1-IIIA | AMP-AM; PIP; TIC; AMC; CIP; LEV; NOR; TMP; TMP-SMX |
| 51144630 LREC-270 | NT:H5 | 1193/53-like | 72142 | 14-64 | mdf(A); sitABCD **gyrA S83L, gyrA D87N, parC S80I, parE L416F** | IncI1-I [ST Unknown] ColBS512-like ColMG828-like | cia, fyuA, gad, iha, irp2, iucC, iutA, kpsE, kpsMII_K1, senB, terC, usp | ISEc31 (iha) | Class I / CAS-Type I/Cas3-0-I; CAS-Type IIIA/Csm2-1-IIIA | CIP; LEV; NOR |
| 51150996 LREC-273 | O75:H5 | 1193/53 | 4085 | 14-64 | mdf(A); sitABCD **gyrA S83L, gyrA D87N, parC S80I, parE L416F** | IncF [F:-A1:B10] ColBS512-like ColMG828-like* Col156-like | chuA, fyuA, gad, iha, irp2, iucC, iutA, kpsE, kpsMII_K1, neuC, ompT, papA_F43, sat, senB, sitA, terC, usp, vat, yfcV | ISEc42 (mdf(A)); MITEEc1 (mdf(A)); MITEEc1 (terC); IS30 (sitABCD, sitA); ISEc31 (iha); IS629 (papA_F43, iutA, sat, iucC) | - | CIP; LEV; NOR |

**TABLE 3** (Continued)

| ID code[a] | O:H antigens[b] | ST#1/ ST#2[c] | cgST[d] | CHType[e] | Acquired resistances and point mutations (in bold)[f] | Plasmid content Inc. group [pMLST][g] | Virulence genes[h] | Mobile genetic elements (& relation to AMR and virulence traits)[i] | CRISPR-Cas system / Cas-type/subtype[j] | Phenotypic AMR[k] |
|---|---|---|---|---|---|---|---|---|---|---|
| 51152710 LREC-275 | O75:H5 | 1193/ 53 | 4085 | 14-64 | bla$_{TEM-1B}$; aph(3'')-Ib, aph(6)-Id; mdf(A), sul2, dfrA14; sitABCD **gyrA S83L, gyrA D87N, parC S80I, parE L416F** | IncF [F-A1:B10] IncQ1 | chuA, fyuA, gad, iha, irp2, iucC, iutA, kpsE, kpsMII_K1, neuC, ompT, papA_F43, sat, senB, sitA, terC, usp, vat, yfcV | ISEc42 (mdf(A)); MITEEc1 (mdf(A)); MITEEc1 (terC); MITEEc1 (irp2; fyuA); IS30 (sitABCD, sitA); ISEc31 (iha); IS629 (papA_F43, iutA, sat, iucC) | Class I / CAS-Type I/Cas3-0-I; CAS-TypeIIIA/Csm2-1-IIIA | AMP-AM; PIP; TIC; CIP; LEV; NOR; TMP; TMP-SMX; FOS |

[a]Isolate and genome (LREC) identification.

[b]O and H antigen prediction with SerotypeFinder 2.0;

[c]Sequence types (ST#1 and ST#2) based on two different MLST schemes were applied: *E. coli* #1 (67) and *E. coli* #2 (22), respectively, and retrieved with MLST 2.0.4.

[d]Core genome ST obtained with cgMLSTFinder1.1. software run against the Enterobase database.

[e]Clonotypes,

[f]acquired antimicrobial resistance genes and/or chromosomal mutations, Resistome: acquired resistance genes: beta-lactam, *bla*$_{TEM-1B}$; aminoglycosides: *aph(3'')-Ib, aph(6)-Id*; macrolides, *mdf(A), mph(A)*; sulphonamides, *sul2*; trimethoprim, *dfrA14, dfrA17*; peroxide, *sitABCD* (mediates transport of iron and manganese and resistance to hydrogen peroxide). Point mutations: quinolones and fluoroquinolones, *gyrA* S83L, TCG-TTG; *gyrA* D87N, GAC-AAC; *parC* S80I, AGC-ATC; *parE* L416F, CTT-TTT.

[g]replicon/plasmid sequence types, Plasmidome: *coverage <100%.

[h]virulence genes, and Virulence determinants: *chuA*, outer membrane hemin receptor; *cia*, colicin Ia; *fyuA*, siderophore receptor; *gad*, glutamate decarboxylase; *iha*, adherence protein; *irp2*, high molecular weight protein 2 non-ribosomal peptide synthetase; *iucC*, aerobactin synthetase; *iutA*, ferric aerobactin receptor; *kpsE*, capsule polysaccharide export inner-membrane protein; *kpsMII_K1*, polysialic acid transport protein group 2 capsule; *neuC*, polysialic acid capsule biosynthesis protein; *ompT*, outer membrane protease (protein protease 7); *papA*_F43, major pilin subunit F43; *sat*, secreted autotransporter toxin; *senB*, plasmid-encoded enterotoxin; *sitA*, iron transport protein; *terC*, tellurium ion resistance protein; *usp*, uropathogenic specific protein; *vat*, vacuolating autotransporter toxin; *yfcV*, fimbrial protein.

[i]the mobile genetic elements associated with AMR and virulence traits were also predicted using: CHtyper 1.0, ResFinder 4.1, PlasmidFinder 2.1, pMLST 2.0, VirulenceFinder 2.0 and MobileElementFinder 1.03 online tools at the Center of Genomic Epidemiology (http://www.genomicepidemiology.org/services/), respectively.

[j]CRISPRCasFinder software (https://crisprcas.i2bc.paris-saclay.fr/) was used to identify and type CRISPR and Cas systems within the genomes.

[k]Phenotypic resistances were interpreted according to EUCAST 2021 breakpoints. Antimicrobial abbreviation: PIP, piperacillin; TIC, ticarcillin; AMC, amoxicillin-clavulanic acid; AMP-AM, ampicillin-amoxicillin; CIP, ciprofloxacin; LEV, levofloxacin; NOR, norfloxacin; TMP, trimethoprim; TMP-SMX, trimethoprim-sulfamethoxazole; FOS, fosfomycin.

purposes. We found one cgST4085 genome (assembly code ESC_FA9684AA_AS) as part of BioProject PRJEB10018 registered in 2016 by the Sanger Institute. We also retrieved two of the three cgST72142 informed as isolates of human origin, Australia, 2007 (ESC_GB6748AA_AS) and 2008 (ESC_TA3850AA_AS) as part of BioProjects PRJNA487890 (submitted by University of Minnesota) and PRJNA493523 (submitted by the University of Queensland), respectively. The *in silico* analysis of these three genomes are summarized in Table S4, and show quite conserved traits regarding acquired resistances, the same set of chromosomal mutations (*gyrA* S83L, *gyrA* D87N, *parC* S80I, *parE* L416F), and a constant in the profile of virulence (*chuA*, *fyuA*, *gad*, *iha*, *irp2*, *iucC*, *iutA*, *kpsE*, *kpsMII* K1, *neuC*, *ompT*, *papA_F43*, *sat*, *senB*, *sitA*, *terC*, *tratT*, *usp*, *vat*, *yfcV*). Besides, PlasmidFinder confirmed the presence of an IncF [F-:A1:B10] plasmid type in the three genomes. The three ST1193 genomes retrieved from Enterobase were also predicted as human pathogen (probability > 92%) when analyzed with PathogenFinder (Table S4).

The genomic similarity of the cgST4085 and cgST72142 assemblies (five from the present study and three from Enterobase) were further investigated through the SNP comparison based on the core genome sequence (82.69% of the reference genome LREC-269) using CSI phylogeny 1.4 (Fig. 1). The phylogenetic dendrogram showed that seven of the eight ST1193 genomes clustered with only 15 to 103 SNP differences. The LREC-270 exhibited the maximum distance, ranging between 140 (in comparison with ESC_TA3850AA) to 204 SNPs (in comparison with LREC-273).

## DISCUSSION

Despite the public health burden implications and derived costs, uUTIs are not systematically monitored, which also means a lack of data regarding the clonal diversity of the *E. coli* involved. Within scientific references, most studies do not differentiate complicated UTI from uUTI, or even from other extraintestinal infections. Likewise, it is not always possible to discriminate the origin of the isolate (hospital or primary health care centers), giving rise to misinterpretations. To avoid this possible bias, we addressed the characterization, using a clonal diagnosis approach, of nonduplicated *E. coli* isolates causing uUTI in women attending primary health care centers associated to the HUCA hospital under a specific pilot study on uUTI named URIPROAS.

It is important to underlined that no set of genes has been found to unequivocally characterize ExPEC and the different categories within a group defined by its isolation from infections outside the intestinal tract. Many cases of bloodstream infections are preceded by an episode of UTI, and therefore UPEC are considered as a subgroup of ExPEC, which also comprise avian pathogenic *E. coli* (APEC) and neonatal meningitis *E. coli* (NMEC) (23). Nevertheless, certain virulence traits have been statistically associated with the pathogenic potential of the isolates, which can be used in a predictive way (24, 25) together with the identification of the so-called global extraintestinal clonal groups of *E. coli* such as ST131 (26). In this study, we used two concepts, namely, the pathogenic categorization and the status of the isolates. Thus, the 100 *E. coli* analyzed here, and clinically recovered from urine as the only pathogen involved in uUTI cases, are UPEC, defined as a subcategory of the ExPEC group. The second concept applied here is the status (UPEC and ExPEC) to differentiate, based on statistical criteria, isolates with higher pathogenic potential. Spurbeck et al. (25) found that *E. coli* isolates carrying *vat*, *fyuA*, *chuA*, *and yfcV* efficiently colonize the urinary tract. Their results indicated that a predictor gene (PG) score of 3 or 4 of those is indicative of status UPEC, and isolates with a PG score of 4 may be highly virulent. In addition, *iutA* and *kpsM II* genes are the most prevalent traits of the status ExPEC scheme (24) for positive isolates, and they are also prevailing in isolates causing extraintestinal infections (18, 27).

The IDSA guidelines recommend trimethoprim-sulfamethoxazole if the local resistance prevalence in *E. coli* is ≤ 20%, and the use of nitrofurantoin and fosfomycin as next-choice antibiotics for uUTI. For pyelonephritis, empirical fluoroquinolone (FQ) treatment is suggested if local resistance prevalence in *E. coli* is ≤ 10% (7). In our study,

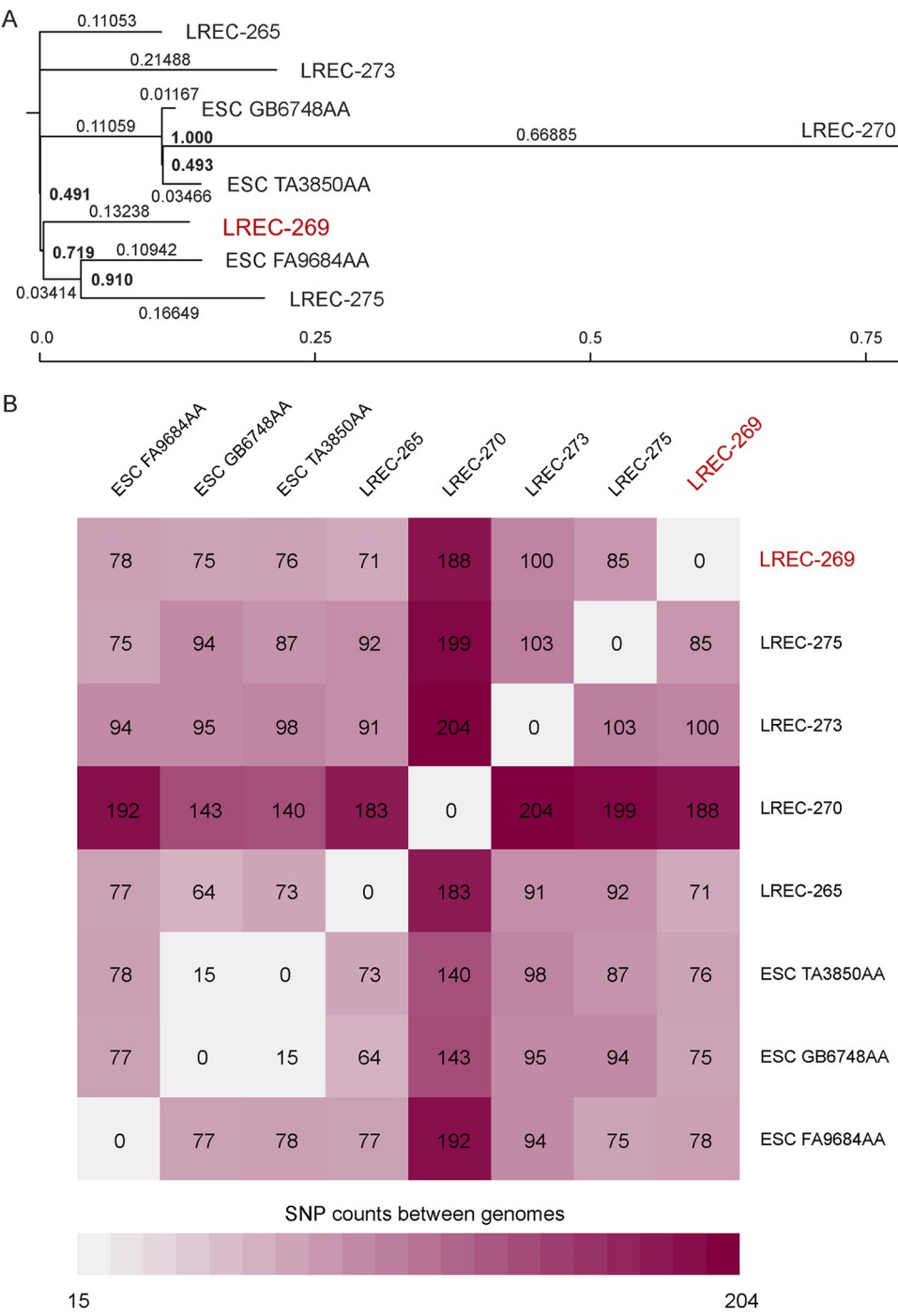

**FIG 1** Phylogenetic dendrogram based on the SNP counts per substitution within the core genome of the ST1193 isolates (five recovered in our study and three retrieved from Enterobase: ESC_FA9684AA_AS, ESC_TA3850AA_AS, ESC_GB6748AA_AS). The comparison of the WGS data sets resulting in a core genome represented by 82.69% of the reference genome (LREC-269). CSI phylogeny version 1.4 (CGE, https://cge.cbs.dtu .dk/services/CSIPhylogeny/; parameters used for phylogenetic analysis: min. depth at SNP positions 10 x; min. relative depth at SNP positions: 10 x; min. distance between SNPs (prune): 10 bp; min. SNP quality: 30; min. read mapping quality: 25, a min. Z-score of 1.96 and by ignoring heterozygous SNPs).

conducted in 2020, the resistance rates observed for fosfomycin (4%) and nitrofurantoin (2%) were similar to those reported for Spain by the ECO·SENS European multicenter study on the antimicrobial resistance of uUTI-associated *E. coli* recovered between 1999 and 2000 (11), and by the ARESC (Antimicrobial Resistance Epidemiological

Survey on Cystitis) in a study conducted between 2003 and 2006 in nine European countries (28). We retrospectively checked resistance rates against the same antibiotics among 241 *E. coli* isolates recovered from women aged 18 to 45 years attending the same primary care centers from August to December of 2020. As a result, we found similar data to those reported for the 100 isolates prospectively studied, which further support the suggestion of fosfomycin and nitrofurantoin as empirical treatment of choice for uUTI by *E. coli* (data not shown). Furthermore, our results corroborate the stable resistance rates ($\sim<4\%$) reported for both antibiotics in specific health areas of Spain such as Galicia, Navarra and Madrid (1999 to 2016) in *E. coli* implicated in UTI (29–31), which further support that these antibiotics should be considered as empirical treatment of choice by the community-acquired uUTI by *E. coli*. In contrast, high percentages of resistance to FQ and trimethoprim-sulfamethoxazole were described for Spain in those same reports (29–31). The variability of non-susceptible isolates to FQ ranged between 10.7% in 2003 to 2006 (28) and 36.6% in 2014 to 2016 (29). For trimethoprim-sulfamethoxazole, resistance ranged between 33.8% of the Spanish isolates within the multicenter European study (2003 to 2006) (28) and 25% of the *E. coli* analyzed in 2016 in Madrid (30). Within our uUTI collection, the resistance rates of trimethoprim-sulfamethoxazole (24%) and FQ (ciprofloxacin: 18%; levofloxacin: 17%; and norfloxacin: 28%) also exceed the limit recommended by the therapeutic guides for the use of these antibiotics as empirical treatments (7). Globally, similar circumstances have been described in different countries for UTIs caused by *E. coli* (uncomplicated or not), where the FQ resistance exceeds even 20%, complicating empirical treatment (18, 32, 33).

Clonotype-level antibiograms can provide a more accurate guidance for empirical treatment than the standard based on species-level antibiograms, as previously showed by Tchesnokova et al. (34, 35) Furthermore, clonal antibiograms could be usable over the time and places due to their stability. But before a clinical implementation, the clonotyping test would benefit from the development of clonal reference databases for the follow up of the dynamics of predominant UTI *E. coli* lineages (32). In the present study, we defined the clonal structure of our collection by means of phylogrouping and clonotyping, followed by MLST for the most prevalent PG-CH combinations. In addition, we performed the screening of specific traits associated with the virulence status of isolates due to the strong correlation found previously between virulence-gene profiles and specific lineages of *E. coli* (18, 27, 36). From the hands-on time and cost point of view, this workflow seems a reasonable approach for the surveillance of emerging clones. As reported in different studies, the phylogenetic group B2 is the predominant within UTI isolates, even though other seven *E. coli* phylogroups *sensu stricto* appear represented to a greater or lesser extent (18, 33, 37, 38). Among the 59 different PG-CH combinations determined in our study, 11 *E. coli* clones comprised 46% of the 100 uUTI isolates. These 11 clones include major pandemic clonal lineages of ExPEC, such as ST73, ST141, ST131, ST69, or ST1193, which are globally responsible for up to 50% of all ExPEC-associated extraintestinal infections worldwide (in particular, UTIs, bloodstream infections, and health care-associated infections) (23, 26, 37, 39).

Our findings can be compared with those resulting from the clonal characterization of 196 consecutively isolated *E. coli* causing UTIs and other extraintestinal infections during 2016 in Spain (100 from Lucus Augusti hospital, HULA, in Lugo) and France (96 from Beaujon hospital in Clichy) (18). Nine out of the 11 uUTI prevalent clones recovered in 2020 in our study were also present within the study collection of Flament-Simon et al. (18): B2-ST73 (CH24-103); B2-ST73 (CH24-30); B2-ST131 (CH40-30); B2-ST141 (CH52-5); B2-ST372 (CH103-9); B2-ST404 (CH14-27); B2-ST1193 (CH14-64); D-ST69 (CH35-27); and F-ST59 (CH32-41). A significant difference found between both studies is the prevalence rate of ST131, which accounted for 12% and 11.5% of the isolates in the Spanish (HULA) and French hospitals, respectively, versus 5% of the present study (health care centers associated to HUCA). In contrast, the MDR rates are quite similar (30% here versus 37.2% for the Spanish and French collection), with individual resistant prevalence $>$ 20% in both studies

for penicillins, FQs, and trimethoprim-sulfamethoxazole. None of the 100 uUTI isolates analyzed in our study were carriers of ESBL enzymes versus 6.6% isolates reported by Flament-Simon et al. (18) for both hospitals, mainly associated to the prevalent ST131. Despite the absence of ESBL-producing ST131 within the uUTI collection, the three B2-ST131 (CH40-30) isolates of our study exhibited MDR, and the one B2-ST131 (CH40-1196) was FQR. It is also important to note that the health area analyzed here, regarding the whole community-acquired and health care-associated UTIs, exhibits ESBL-producing *E. coli* rates of around 10% (unpublished data). This fact corroborates that despite the global dispersion of *E. coli* in the community, ESBL-producing isolates are commonly related to patients with risk factors such as previous hospitalization and chronic diseases, which are normally rare in young women (40, 41). To summarize, and taking into consideration the miscellaneous nature of the extraintestinal isolates analyzed in the study of Flament-Simon et al. (18) and the different year of isolation (2016), the similarities found suggest that clonal and antibiotic susceptibility surveillance is essential at both local and global level to evaluate the evolutive impact of the antibiotic use on UTIs as a whole (complicated and uncomplicated).

Over the past 2 decades, the emergence of FQR *E. coli* clones such as B2-ST131 (CH40-30), F-ST648 (CH4-27), or B2-ST1193 (CH14-64) has widely contributed to the spread of MDR and treatment failure (42–44). Particularly B2-ST1193 (CH14-64) has been described since 2012 as a newly emerging ExPEC clone, and an important contributor to the FQR within the *E. coli* population across several countries, i.e., France (45), Germany (46), China (47–50), Korea (51), Vietnam (52) and the United States (16, 53). An impacting finding of a study performed on fecal and urine samples from women without documented UTI (2015 to 2016) was the presence of FQR *E. coli* in 8.8% of healthy women, with most bacteria belonging to the pandemic ST131-*H30R* or ST1193 clonal groups (54). According to the authors, these uropathogenic FQR clonal groups have enhanced ability to persist in the gut and cause bacteriuria in healthy women. To date, most ST1193 have been recovered from human samples generally associated with bloodstream infections, UTIs or meningitis, and with limited overlap with animal sources (52, 55). It is also displayed in Enterobase (https://enterobase.warwick.ac.uk/), which includes a total of 1,199 ST1193 genomes (December 19, 2021), almost all referred as human origin and only a few recovered from wildlife (seagull), food, bovine, companion animal, or environment.

In the present study, we found an outstanding occurrence of FQR ST1193 isolates (6%) in uUTI. In contrast, Flament-Simon et al. (18) had found a prevalence of 1% among each of the Spanish and French hospital isolates recovered in 2016.

The genomic analysis of our typically lactose negative ST1193 isolates showed similar key genomic characteristics than global ST1193 (56, 57). Thus, the Spanish uUTI isolates were positive for the K1 capsular type, exhibited mutations in gyrase A (g*yrA* D87N and *gyrA* S83L) and DNA topoisomerase IV genes (*parC* S80I and *parE* L416F). An IncF [F-:A1:B10] plasmid type was also common (with the exception of LREC-270) together with other Col-like and/or plasmid Inc. types. We found almost identical virulence profile of traits commonly prevalent in ST1193 (*chuA, fyuA, iha, irp2, iucC, iutA, kpsE, kpsMII, neuC, ompT, sat, senB, sitA, terC, usp, vat, yfcV*). Among those, *senB* is a plasmid-carried enterotoxin gene associated with enhancing bladder colonization and invasion (58). We confirmed that *senB* was present in all our isolates with plasmid assignation. It has been suggested that this virulence trait presence shown by many high-risk ExPEC clones, together with the high prevalence of a IncF [F-:A1:B10] plasmid in *E. coli* ST1193, might be implicated in the epidemiologic spread of this clone (59). None of our isolates were ESBL-producers; however, it is of concern the increasing presence in other countries of ESBL-producing ST1193 isolates with a wide variety of $bla_{CTX-Ms}$, including CTX-M-15, CTX-M-27, or CTX-M-55 (55, 57, 59, 60). These studies suggest a transition in sequence type (ST) and in prevalence of ESBL types, as well as a change in the epidemiology of UPEC isolates that must be followed up. Contrary to previous

studies, we found CRISPR arrays in three of our isolates (59, 61), which exhibited the same CRISPR-cas system.

To investigate the phylogeny of our isolates in a global context, we used genomic data based on the cgMLST scheme and the hierarchical clustering scheme (HierCC) of Enterobase to retrieve three genomes belonging to the cgSTs 4085 and 72142. HierCC, based on core genome multi-locus typing, allows incremental, static, multilevel cluster assignments of genomes (62, 63). Most of the ST1193 genomes displayed in Enterobase (1,178 of 1,199) are assigned into the same HierCC HC50 (26), which means all genomes in this cluster have links no more than 50 alleles apart. Furthermore, ESC_FA9684AA_AS (cgST4085), ESC_TA3850AA_AS, and ESC_GB6748AA_AS (cgST72142) cluster together in HC20 (571) with no more than 20 alleles of difference. The SNP-based phylogenetic tree comparison confirmed the high core genome similarity between the eight genomes (the three from Enterobase and five of the present study). According to the metadata uploaded in Enterobase, this result would prove the fact of a conserved ST1193 clone circulating in different countries and years. As proposed by Zhou et al. (63), a real-time surveillance would benefit from immediate assignments of each genome assembly to hierarchical population structures such as the HierCC scheme of Enterobase.

In conclusion, the present study provides evidence of the importance of urine culture for the treatment for uUTIs based on susceptibility patterns of the isolates. We suggest here that clonal diagnosis, antimicrobial susceptibility testing (AST) and the screening of predictor genes could help in monitoring the risk of progression to a more severe condition, at the individual level, and globally, in the monitoring of UTIs. We also highlight the urgent need of surveillance to detect changes in the dynamics of emerging UPEC clones associated with the spread of FQR, such as the pandemic ST1193 clonal group, which has the potential to acquire additional resistance genes and colonize healthy individuals. Furthermore, our study represents the first report of ST1193 in uUTI in Spain.

## MATERIALS AND METHODS

**E. coli collection.** We conducted a prospective, multicenter, specific pilot study on uUTI named URIPROAS. This study included 100 nonduplicated *E. coli* isolates recovered from urine samples collected in March ($n = 45$) and June/July ($n = 55$) of 2020 from women between 18 and 45 years (median age 34) attending 15 primary health care centers associated to the Hospital Universitario Central de Asturias (HUCA), northern Spain (Table S1). To avoid bias, the first 100 isolates recovered during those periods of time (45 and 55, respectively) were included in the study. The health area of the HUCA hospital serves a population of approximately 330,000.

Bacterial identification was performed by matrix-assisted laser desorption/ionization–time-of-flight mass spectrometry (MALDI-TOF) (Bruker Daltonik, Bremen, Germany) after conventional culture. A reliable result (at the species level) was only considered if the score obtained was higher than 2. The selected isolates were stored at room temperature in nutrient broth (Difco) with 0.75% nutrient agar (Difco) for further characterization in the Reference Laboratory for *E. coli* (LREC-USC) (Table S1).

**PCR screening of virulence traits.** The 100 UPEC isolates recovered from uUTIs were screened by PCR for specific marker traits associated with the status of virulence. Thus, the purpose of the screening was not an exhaustive characterization, but the subtyping of *E. coli* isolates causing the same type of infection (uUTI), bearing in mind that those traits were statistically associated with higher potential pathogenicity and prevalence in previous studies (24, 25, 27). Specifically, the status of uropathogenic *E. coli* (UPEC) was assigned to those isolates positive for $\geq 3$ of the following marker genes (*chuA, fyuA, vat,* and *yfcV*) (25). While for the status of extraintestinal *E. coli* (ExPEC), a duplex PCR based on *iutA* (441 bp) and *kpsMII* (272 bp) genes was performed as described elsewhere (27).

**Phylogroup, clonotype, and sequence type assignment.** The clonal structure of the *E. coli* collection was established by means of phylogrouping, clonotyping and MLST (64–67). Briefly, the phylogroup of the 100 isolates was established according to the PCR-based method developed by Clermont et al. (64, 65), relying on the combination of the presence of four DNA markers followed by allele-specific amplifications, which allows the rapid identification of the eight *E. coli* phylogroups belonging to *E. coli sensu stricto* (A, B1, B2, C, D, E, F, G). The clonotyping was performed by sequencing the internal 469-nucleotide (nt) and 489-nt sequence of the *fumC* (allele obtained from MLST) and *fimH* genes, respectively (66). The MLST was conducted for those isolates belonging to the most prevalent phylogroup-CH combinations ($\geq 3$ isolates), following the Achtman scheme of seven genes (67). The ST was assigned through the EnteroBase website (http://mlst.warwick.ac.uk/mlst/dbs/Ecoli).

**Antimicrobial susceptibility testing.** MICs of the 100 *E. coli* isolates were determined using the MicroScan WalkAway System (Beckman Coulter, Brea, CA, USA). A total of 28 drugs were included in the analysis: penicillins (ampicillin/amoxicillin, piperacillin, ticarcillin); antipseudomonal penicillins + beta-lactamase inhibitors

(piperacillin-tazobactam); penicillins + beta-lactamase inhibitors (amoxicillin-clavulanic acid); non-broad spectrum cephalosporins (cefuroxime); broad-spectrum cephalosporins (cefixime, cefotaxime, ceftazidime, cefepime); broad-spectrum cephalosporins + beta-lactamase inhibitors (ceftolozane-tazobactam, ceftazidime-avibactam); carbapenems (imipenem, ertapenem, meropenem); monobactams (aztreonam); fluoroquinolones (norfloxacin, ciprofloxacin, levofloxacin); aminoglycosides (gentamicin, tobramycin, amikacin); glycylcyclines (tigecycline); nitrofurans (nitrofurantoin); phosphonic acids (fosfomycin); folate pathway inhibitors (trimethoprim, trimethoprim/sulfamethoxazole); and polymyxins (colistin). Additionally, colistin susceptibility was analyzed with the standard broth microdilution (BMD) method as described elsewhere (68). Susceptibility testing was interpreted according to EUCAST 2021 breakpoints. Isolates were classified as MDR if displayed resistance to a drug of $\geq$ three or more of the above-mentioned antimicrobial categories.

**Whole genome sequencing.** Isolates belonging to ST1193 were further investigated by WGS. The genomic DNA libraries for sequencing were prepared using the Nextera XT Library Prep kit (Illumina, CA, USA) according to the manufacturer's recommendation. Libraries were purified using the Mag-Bind RXNPure Plus magnetic beads (Omega Biotek), following the instructions provided by the manufacturer. Then, libraries were pooled in equimolar amounts according to the quantification data provided by the Qubit dsDNA HS Assay (Thermo Fisher Scientific). Lastly, the libraries were sequenced in an Illumina NovaSeq PE150 platform, obtaining 100 to 150 bp paired-end reads which were trimmed (Trim Galore 0.6.0) and filtered according to quality criteria (FastQC 0.11.9). The quality-filtered reads were assembled *de novo* using Unicycler (v0.4.8) (69), which uses an adapted SPAdes (v3.14.0) assembling algorithm (70). For the comprehensive typing of the isolates, the assembled contigs were analyzed using different bioinformatics tools of the Center for Genomic Epidemiology (CGE) as specified, and applying the thresholds suggested by default when required (minimum identity of 95% and coverage of 60%): SeroTypeFinder 2.0 (71), CHTyper 1.0 (72), MobileElementFinder 1.03 (73), PlasmidFinder 2.1 and pMLST 2.0 (74). For the phylogenetic typing, two different MLST schemes were applied: *E. coli* #1 (67) and *E. coli* #2 (22). Additionally, cgMLSTFinder1.1 (75) was applied for the core genome multi-locus typing (cgMLST) from the raw reads of the isolates. cgMLSTFinder software runs KMA (k-mer alignment) against the Enterobase (core genome MLST, cgMLST) database, based on a total number of 2,513 conserved loci analyzed (62). In addition to this typing, lineage-specific gene markers based on the sequence of CRISPRCasFinder software (https://crisprcas.i2bc.paris-saclay.fr/) was used to identify and type CRISPR and Cas systems within the genomes.

Two different tools were used for the identification of acquired genes and/or chromosomal mutations mediating antimicrobial resistance, ResFinder 4.1 (76–78) first, and then ABRIcate v1.0.1 (https://github.com/tseemann/abricate) was run against the curated CARD database (https://card.mcmaster.ca/home). The identification of acquired virulence genes was performed using the web-based tool VirulenceFinder 2.0 (79, 80), and then ABRIcate v1.0.1 was run against the virulence factor database (VFDB) (81) where results were filtered only for *E. coli* entries. Besides, the bacteria's pathogenicity toward human hosts was predicted using PathogenFinder 1.1 (82).

A phylogeny based on the concatenated alignment of the high-quality SNPs was inferred using CSI phylogeny 1.4 (Call SNPs & Infer Phylogeny) as described (83). Enterobase (https://enterobase.warwick.ac.uk/) was searched for genomes belonging to ST1193 (according to the Achtman 7-gene MLST scheme) and specific core genome MLST (cgMLST) sequences, whose assemblies were retrieved for comparative purposes.

**Statistical analysis.** Comparisons of proportions were tested using a two-tailed Fisher's exact test. $P$ values $<$ 0.05 were considered statistically significant.

**Data availability.** The nucleotide sequence of the ST1193 LREC isolates were deposited in the European Nucleotide Archive (ENA) with the following accession (ERR7926813 to ERR7926817) and BioSample (SAMEA11224177 to SAMEA11224181) codes, as part of BioProject ID PRJEB49681.

## SUPPLEMENTAL MATERIAL

Supplemental material is available online only.
**SUPPLEMENTAL FILE 1**, XLSX file, 0.03 MB.
**SUPPLEMENTAL FILE 2**, XLSX file, 0.03 MB.
**SUPPLEMENTAL FILE 3**, XLSX file, 0.04 MB.
**SUPPLEMENTAL FILE 4**, XLSX file, 0.01 MB.

## ACKNOWLEDGMENTS

This study was supported by the projects and funds PID2019-104439RB-C21/AEI/10.13039/501100011033 from the Agencia Estatal de Investigación (AEI, Spain), co-funded by the European Regional Development Fund of the European Union: A Way to Make Europe (ERDF); FIS PI17-00728 (Fondo de Investigación Sanitaria, Instituto de Salud Carlos III, Ministerio de Economía y Competitividad, Spain), co-funded by ERDF; GRUPIN IDI/2022/000033 by the Regional Ministry of Science of Asturias (IDI/2022/000033). ED431C 2021/11 from the Consellería de Cultura, Educación e Ordenación Universitaria (Xunta de Galicia) and ERDF. I.G-M. and V.G. acknowledge the Consellería de Cultura, Educación e Ordenación Universitaria, Xunta de Galicia for their postdoctoral grants (Grant Number ED481B-2021-006 and ED481-B2018/018, respectively).

The research stay of I.G-M at the Hospital Universitario Central de Asturias was funded by a grant from the Sociedad Española de Enfermedades Infecciosas y Microbiología Clínica (SEIMC). L.L. acknowledges the Ministry of Education of Spain for her predoctoral grant FPU19/01127.

The present study was approved by the ethics committee of the Principality of Asturias within the framework of the study URIPROAS.

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
