## [Reviewer comments · Microbiology Spectrum]

Microbiology Spectrum

Occurrence and genomic characterization of clone ST1193 clonotype (CH)14-64 in uncomplicated urinary tract infections caused by *Escherichia coli* in Spain

Isidro García-Meniño, Pilar Lumbreras, Luz Lestón, Mónica Alonso, Vanesa García, Jens Hammerl, Javier Fernández, and Azucena Mora

Corresponding Author(s): Azucena Mora, Universidade de Santiago de Compostela

Review Timeline:

Submission Date:	January 6, 2022
Editorial Decision:	February 14, 2022
Revision Received:	March 31, 2022
Accepted:	April 26, 2022

Editor: Sadjia Bekal

Reviewer(s): The reviewers have opted to remain anonymous.

Transaction Report:

DOI: <https://doi.org/10.1128/spectrum.00041-22>

February 14, 2022

Dr. Azucena Mora
Universidade de Santiago de Compostela
Departamento de Microbioloxía e Parasitoloxía
Laboratorio de Referencia de E. coli (LREC)
Facultade de Veterinaria
Lugo 27002
Spain

Re: Spectrum00041-22 (Occurrence and genomic characterization of clone ST1193 clonotype (CH)14-64 in uncomplicated urinary tract infections caused by Escherichia coli in Spain)

Dear Dr. Azucena Mora:

Link Not Available

Sincerely,

Sadjia Bekal

Journals Department
Reviewer comments:

Reviewer #1 (Comments for the Author):

The paper by García-Meniño et al. presents a punctual analysis of E. Coli strains from uncomplicated UTI cases in pre-menopausal women from a region in Spain. 100 distinct isolates of E. coli were examined using different typing schemes, determination of antibiotic resistance, and screening for a few virulence factors known to be described as representative of UPEC or ExPEC strains.

The work is relevant for surveillance as there have been less studies focusing on uncomplicated community acquired strains and it is important to determine if certain specific clones are commonly causing such infections as well as what types of antibiotic resistance are more prevalent. This is important, as it provides a basis for the choice of antibiotic treatment to prescribe. Based on the results presented, the authors consider nitrofurantoin and fosfomycin to be the best course of treatment as there was a low level of resistance among strains. However, although 100 isolates is a considerable number, it would help re-enforce whether the frequency of resistance of these two antibiotics among a greater number of strains. The identification of the specific clonal groups and presence of ST1193 as a more commonly identified group in the uUTI isolates is also an interesting aspect, as well as the sequencing and genomic comparison of those strains.

Overall the paper is clearly written, although there are a few aspects that require revision or justification.

The results from the use of the "UPEC" or "ExPEC" -specific virulence screens are of limited use and demonstrate that there may commonly be E. coli strains that may not fall into the UPEC or ExPEC categorization using the screening for presence of a few genes.

The bias of limiting selection of strains to be UPEC due to chuA, fyuA, yfcV, vat and kps and iutA as ExPEC is limiting, as it would eliminate potentially different groups of UPEC that may possess distinct sets of virulence genes that could contribute to their pathogenic potential in UTI.

Line 163- Did the 31% (nearly 1/3) that did not fall into the UPEC category have other attributes indicating they may be UPEC/ExPEC associated with an infection or may represent possibly E. coli that contaminated a urine sample or were not likely to be the cause of a UTI? It would be important to know if these isolates considered as non-UPEC or non-ExPEC based on the screening methods were shown to be part of some of the more predominant clones associated with uUTIs.

The use of these limited sequence screening criteria, despite being published by others, for characterizing a strain as ExPEC or UPEC are very limited and to an extent can be inaccurate. By definition, all UPEC are ExPEC which is a general term for a larger set of isolates of E. coli causing a variety of systemic infectious disease... meningitis, wound infections, sepsis, as well as UTIs. Using the limited gene screens, the results indicate only 44% are ExPEC, but 69% are UPEC? Based on this comparison 25% of the UPEC are considered not to be ExPEC? This is clearly not true based on the other types of analyses. They are likely to be ExPEC as well and likely contain other ExPEC virulence-associated genes. Clearly there can be many ExPEC that do not necessarily contain the kps and iutA genes.... It would be important to include this type of information in the discussion, to at least acknowledge that these screening criteria have limitations and may miss important subgroups of E. coli, that may not have these sequences, but nevertheless can cause UTIs or other types of extra intestinal infections.

Line 173- the fact that 65 of 67 of the B2 strains were "UPEC" based on the sequence screening further shows the choice of "virulence genes" are more closely linked to B2 phylogroup and not only UTI or ExPEC associated virulence factors...

Line 301- The detailed inclusion of the specific clonal groups and numbers from a previous study, listing all of them is very detailed and atypical for the discussion. It would be better to just overview similarities and not delve into highly detailed comparisons in the discussion.

Much of the discussion restates what has been presented in the results. It would be of more interest to the readers to relate the findings of the study to those of other reports or describe what other studies have commonly identified that were also found in the current work. It would also be of value to include discussion concerning the discrepancies found between the UPEC/ExPEC gene screening which only indicated a likelihood of 2/3 of the strains to be either ExPEC or UPEC, and that there may be some UPEC clones that lack the sequences used in these simplified gene screens of only a few specific genes commonly linked mainly to B2 isolates.

Although accession numbers for genomes and sequence data are given in the text, the data has not been released for the accession numbers. Such sequence data should normally be available/accessible during the review process. In any case, this data should definitely be made immediately available prior to publication if the paper is accepted.

Minor corrections:

Line 75- replace "within this" with "to treat these", otherwise something is wrong or missing in this sentence.

Line 250- should be "differentiate complicated UTI from uUTI"

Line 252- should be giving and not given.

Line 372- "importance" would be a more appropriate and accurate word than "necessity"

Line 375- change to "pandemic ST1193 clonal group, which .."

Reviewer #3 (Comments for the Author):

This study characterized 100 E. coli isolates from uncomplicated urinary tract infections from one health region in Spain. Various methods of strain typing were performed along with antimicrobial susceptibility testing. Five isolates belonging to ST1193 CH14-64 were subjected to whole genome sequencing and further analysis.

Major comments

The study provided information to inform the local antibiogram of E. coli from uUTI in this particular health region. This is useful information since the susceptibility pattern suggests that second line recommended therapies are more appropriate than the first line recommendation. This conclusion was consistent with several other larger studies that were cited. Meaningful conclusions cannot be drawn from the WGS portion of the manuscript as only 5 isolates were sequenced. Given that there were only 3 isolates in Enterobase matching this cgMLST (out of 1900 isolates of ST1198), this does not seem to be one of the prevalent strain types causing uUTIs in Spain or globally. Analysis of all isolates that were resistant to first line agents would be more informative. The discussion section was very thorough.

Minor Comments

Line 18-19. Specify that the 15 centres were from one health region.

Abstract: suggest highlighting high diversity of E. coli causing uUTI

Line 80-81: if there was no potential selection bias, specify e.g. if you selected all samples in March and first 55 in June/July.

Line 100: briefly state what the method of Clermont is based on.

Line 110-111: I think amox/clav and pip/tazo are generally considered to part of the same drug class of penicillin + beta-lactamase inhibitor.

Line 166: Is it more correct to say that, "44% of E. coli were carriers of ONE OR both iutA and kpsMII"?

Staff Comments:

Preparing Revision Guidelines

Please return the manuscript within 60 days; if you cannot complete the modification within this time period, please contact me. If you do not wish to modify the manuscript and prefer to submit it to another journal, please notify me of your decision immediately so that the manuscript may be formally withdrawn from consideration by Microbiology Spectrum.

Dear Editor and reviewers,

We want to thank you for the comments and suggestions, which have helped us to rewrite the second version of this manuscript. Please, find below our point-by-point answers to the issues raised. We hope the manuscript is now suitable for acceptance.

Reviewer comments:

Reviewer #1 (Comments for the Author):

A) The paper by García-Meniño et al. presents a punctual analysis of *E. coli* strains from uncomplicated UTI cases in pre-menopausal women from a region in Spain. 100 distinct isolates of *E. coli* were examined using different typing schemes, determination of antibiotic resistance, and screening for a few virulence factors known to be described as representative of UPEC or ExPEC strains.

The work is relevant for surveillance as there have been less studies focusing on uncomplicated community acquired strains and it is important to determine if certain specific clones are commonly causing such infections as well as what types of antibiotic resistance are more prevalent. This is important, as it provides a basis for the choice of antibiotic treatment to prescribe. Based on the results presented, the authors consider nitrofurantoin and fosfomicin to be the best course of treatment as there was a low level of resistance among strains. However, although 100 isolates is a considerable number, it would help re-enforce whether the frequency of resistance of these two antibiotics among a greater number of strains.

The identification of the specific clonal groups and presence of ST1193 as a more commonly identified group in the uUTI isolates is also an interesting aspect, as well as the sequencing and genomic comparison of those strains.

Firstly, we want to thank the reviewer's comments highlighting the relevance of the study.

We agree with reviewer 1 that a greater number of strains would reinforce fosfomicin and nitrofurantoin as empirical treatment of choice for uUTI by *E. coli*. Nevertheless, our results are consistent with the stable resistance rates (~<4%) observed in different European studies for both antibiotics, as well as in specific health areas of Spain, which further support these antibiotics as empirical treatment of choice by the community-acquired uUTI by *E. coli*. **This explanation is already included in the discussion section (Lines 219-222).**

To support our findings, **we have retrospectively checked resistance rates against the same antibiotics among 241 *E. coli* isolates recovered from women aged 18 to 45 years attending the same primary care centers from August to the December of 2020. We have found similar data to those reported here for the 100 isolates prospectively studied**, which reinforces the suggestion of fosfomicin and nitrofurantoin as empirical treatment of choice for uUTI by *E. coli*. **We have included this information in the Discussion section (Lines 214-219)**

B) Overall, the paper is clearly written, although there are a few aspects that require revision or justification.

Thank you for the comment.

C) The results from the use of the "UPEC" or "ExPEC" -specific virulence screens are of limited use and demonstrate that there may commonly be *E. coli* strains that may not fall into the UPEC or ExPEC **categorization** using the screening for presence of a few genes.

We would like to clarify first that **in the present study, we are using two concepts**. Namely, the pathogenic categorization, and the status of the isolates.

-The categorization: the collection of study here, unequivocally belongs to the uropathogenic *E. coli* (UPEC) subgroup, which is part of the extraintestinal pathogenic *E. coli* (ExPEC) group.

-The status: meaning the pathogenic capacity of causing severe disease, and defined on the bases of certain virulence traits, statistically associated to higher potential pathogenicity (status ExPEC, status UPEC).

D) The bias of limiting selection of strains to be UPEC due to *chuA*, *fyuA*, *yfcV*, *vat* and *kps* and *iutA* as ExPEC is limiting, as it would eliminate potentially different groups of UPEC that may possess distinct sets of virulence genes that could contribute to their pathogenic potential in UTI.

Line 163- Did the 31% (nearly 1/3) that did not fall into the UPEC category have other attributes indicating they may be UPEC/ExPEC associated with an infection or may represent possibly *E. coli* that contaminated a urine sample or were not likely to be the cause of a UTI?

Following with the previous answer, it is true that no set of genes has been found to unequivocally characterise extraintestinal pathogenic *E. coli* (ExPEC) and the different categories within the ExPEC group, which is defined by its isolation from infections outside the intestinal tract (Riley, 2020).

All the isolates included in the present study, and clinically recovered from urine as the only pathogen isolated in these uUTI cases, are uropathogenic *E. coli* (UPEC), defined as a subcategory of the Extraintestinal pathogenic *E. coli* group (ExPEC).

The second concept applied here is the UPEC and ExPEC status to differentiate, based on statistical criteria, isolates **with higher pathogenic potential**. Spurbeck et al (2012) found that ***Escherichia coli* isolates that carry *vat*, *fyuA*, *chuA*, and *yfcV* efficiently colonize the urinary tract**. As stated above, since ExPEC are a heterogeneous group of pathogens that encompasses avian, neonatal meningitis, and uropathogenic *E. coli* strains, and there were no core set of virulence factors that can be used to definitively differentiate these pathotypes, the authors analyzed four virulence factor-encoding genes, *yfcV*, *vat*, *fyuA*, and *chuA*, highly associated with uropathogenic *E. coli* strains. **Their results indicated that a predictor gene (PG) score of 3 or 4 of those is indicative of status UPEC, and isolates with a PG score of 4 may be highly virulent.**

In summary, the purpose of the PCR screening of a limited number of specific traits was not an exhaustive characterization, but the subtyping of *E. coli* isolates causing the same type of infection (uUTI), bearing in mind that those traits were statistically associated with higher potential pathogenicity and prevalence in previous studies.

To make this clearer, we have included the above information in the Material and Methods section: **"PCR screening of virulence traits. The 100 UPEC recovered from uUTIs were screened by PCR for specific marker traits associated with the status of virulence. Thus, the purpose of the screening was not an exhaustive characterization, but the subtyping of *E. coli* isolates causing the same type of infection (uUTI), bearing in mind that those traits were**

statistically associated with higher potential pathogenicity and prevalence in previous studies” (Lines 349-354).

Also, it has been added in the Result: **“Virulence status. The screening of virulence traits associated with a higher efficiency in the colonization of the urinary tract revealed that 69 of the 100 uUTI isolates conformed the UPEC status (≥ 3 of specific virulence traits *chuA*, *fyuA*, *vat* and *yfcV*). Individually, *chuA*, *fyuA*, *vat* and *yfcV* were present in 91%, 91%, 59% and 67% of the isolates, respectively.” (Lines 89-93).**

In addition, *iutA* and *kpsM II* genes are the most prevalent traits of the ExPEC status scheme for positive isolates, and they are also prevailing in isolates causing extraintestinal infections (Díaz-Jiménez et al. 2021; Flament-Simon et al., 2020). This information has also been included **“the screening of *iutA* and *kpsMII* as prevailing genes within extraintestinal infections, showed that 44% of the *E. coli* were carriers of both.” (Lines 93-94).**

In fact, we believe that **it could be suggested the screening of these reduced number of virulence genes as a simple scheme in the routine diagnosis workflow of the laboratory with the purpose of subtyping *E. coli* isolates causing the same type of infection (uUTI).** This suggestion has been included in the new version (Lines 326-328).

E) It would be important to know if these isolates considered as non-UPEC or non-ExPEC based on the screening methods were shown to be part of some of the more predominant clones associated with uUTIs.

Once it is clarified that the 100 isolates are UPEC, within the ExPEC category; it is important to highlight the high diversity of clones represented by 59 different phylogroup-clonotype combinations. This diversity is in line with previous reports on UTI (Tchesnokova et al., 2013; Matsumura et al., 2017; Flament-Simon et al., 2020). However, and also in line with previous studies, there are certain predominant UPEC clones. These predominant clones show specific traits of resistance and/or virulence.

Almost 70% of our isolates satisfied the UPEC status, including globally predominant clones such as B2-ST131 (CH40-30), B2-ST141 (CH52-5), B2-ST1193 (CH14-64) or B2-ST73 (CH24-30). **Regarding the isolates not conforming the UPEC status (31 isolates), most of them belonged to clones represented by a single isolate carrying non-predominant clonotypes, except for D-ST349 (CH-54) and D-ST69 (CH35-27) (4 and 5 isolates, respectively). D-ST69 (CH35-27) is a successful clone associated with MDR and typically does not conform UPEC status** (Tchesnokova et al., 2013; Flament-Simon et al., 2020).

Now, we have included the following explanation in the Result: **“Most isolates (36 of 46) belonging to 9 of the 11 prevalent clones were carriers of ≥ 3 of specific virulence traits *chuA*, *fyuA*, *vat* and *yfcV* associated with higher virulence, and therefore, positive for the UPEC status; and only clones D-ST349 (CH-54) and D-ST69 (CH35-27) were characteristically negative. Regarding the presence or absence of both *iutA* and *kpsMII*, we observed a positive correlation with isolates belonging to clones B2-ST1193 (CH14-64), B2-ST73 (CH24-30), B2-ST404 (CH14-27), B2-ST131 (CH40-30) and F-ST59 (CH32-41).” (Lines 111-116).**

F) The use of these limited sequence screening criteria, despite being published by others, for characterizing a strain as ExPEC or UPEC are very limited and to an extent can be inaccurate. By definition, all UPEC are ExPEC which is a general term for a larger set of isolates of *E. coli* causing a variety of systemic infectious disease... meningitis, wound infections, sepsis, as well

as UTIs. Using the limited gene screens, the results indicate only 44% are ExPEC, but 69% are UPEC? Based on this comparison 25% of the UPEC are considered not to be ExPEC? This is clearly not true based on the other types of analyses. They are likely to be ExPEC as well and likely contain other ExPEC virulence associated genes. Clearly there can be many ExPEC that do not necessarily contain the *kps* and *iutA* genes.... It would be important to include this type of information in the discussion, to at least acknowledge that these screening criteria have limitations and may miss important subgroups of *E. coli*, that may not have these sequences, but nevertheless can cause UTIs or other types of extra intestinal infections.

Please, see answers **A** to **E**.

In the new version of our manuscript, we have explained from the beginning the concepts of categorization and status.

We have also added in the Discussion section the following information (Lines 190-206). ***“It is important to underline that no set of genes has been found to unequivocally characterise ExPEC and the different categories within a group defined by its isolation from infections outside the intestinal tract. Many cases of bloodstream infections are preceded by an episode of UTI, and therefore UPEC are considered as a subgroup of ExPEC, which also comprise avian pathogenic *E. coli* (APEC) and neonatal meningitis *E. coli* (NMEC) (Riley, 2014). Nevertheless, certain virulence traits have been statistically associated with the pathogenic potential of the isolates, which can be used in a predictive way (Johnson et al., 2003; Spurbeck et al., 2012) together with the identification of the so-called global extraintestinal clonal groups of *E. coli* such as ST131 (Manges et al., 2019). In this study, we used two concepts, namely, the pathogenic categorization and the status of the isolates. Thus, the 100 *E. coli* analyzed here, and clinically recovered from urine as the only pathogen involved in uUTI cases, are UPEC, defined as a subcategory of the ExPEC group. The second concept applied here is the status (UPEC and ExPEC) to differentiate, based on statistical criteria, isolates with higher pathogenic potential. Spurbeck et al (2012 doi: 10.1128/IAI.00752-12) found that *E. coli* isolates carrying *vat*, *fyuA*, *chuA*, and *yfcV* efficiently colonize the urinary tract. Their results indicated that a predictor gene (PG) score of 3 or 4 of those is indicative of status UPEC, and isolates with a PG score of 4 may be highly virulent. In addition, *iutA* and *kpsM II* genes are the most prevalent traits of the status ExPEC scheme (Johnson et al., 2003) for positive isolates, and they are also prevailing in isolates causing extraintestinal infections (Díaz-Jiménez et al. 2021; Flament-Simon et al., 2020).”***

G) Line 173- the fact that 65 of 67 of the B2 strains were "UPEC" based on the sequence screening further shows the choice of "virulence genes" are more closely linked to B2 phylogroup and not only UTI or ExPEC associated virulence factors...

It seems interesting to highlight that **not only B2 isolates of specific STs** are associated with the carriage of those virulence traits, **but also F and G specific lineages of *E. coli***. We probed previously, in a comprehensive study on the recovery and characterization of *E. coli* from poultry meat to analyze the microbiological risk for consumers, that the duplex PCR based on *iutA* and *KpsM II* genes was essential for the accurate screening of the isolates with ExPEC status, as well as for the recovery of those with UPEC status, since most of the latter also satisfies the ExPEC status (but not the other way around) (Díaz-Jiménez et al. 2021). Eight-three isolates out of 323 *E. coli* recovered from 100 samples conformed the UPEC status, which belonged to B2 (68.7%), F (25.3%) and G (6%). Within the 22 STs established for the 83 meat

isolates, we found some of the most prevalent in UPEC human collections, such as ST95-B2, ST131-B2 and ST141-B2. The relevant presence of isolates belonging to phylogroups F and G within poultry meat was mostly due to the clones ST648-F (CH4-58), ST1485-F (CH231-58) and ST117-G (CH45-97), which were also in the human clinic collection, but especially within the ESBL-producing *E. coli* (Flament-Simon et al., 2020a; Flament-Simon et al., 2020b)

This strong correlation between virulence-gene profiles and STs (and, by extension, with phylogroups) has been reported previously in human clinical isolates, as stated above. Thus, all B2-ST12, B2-ST73, B2-ST95, B2-ST127 and B2-ST141 isolates and most B2-ST131 isolates showed the status UPEC. In contrast, none of the A-ST10, B1-ST58, D-ST69 and C-ST88 isolates presented the VF necessary to satisfy the UPEC status. **Likewise, specific lineages of *E. coli* are commonly MDR** such as B2-ST131 (CH40-30), B2-ST69 (CH35-27), C-ST88 (CH4-39) or the emerging B2-ST1193 (CH14-64) (Flament-Simon et al., 2020a). All these results suggest that clonal and antibiotic susceptibility surveillance is essential at both local and global level to evaluate the evolutive impact of the antibiotic use.

In this new version of the manuscript, we have rewritten our findings in the Results as follows ***“The predictor genes associated with a higher efficiency in the colonization of the urinary tract, and therefore positive for the UPEC status was found within the isolates belonging to phylogroups B2 (65 of 67 isolates), F (3) and G (1). Both iutA and kpsMII virulence traits were present in 44 isolates of phylogroups A (2 out of 5 isolates), B2 (32 of 67), D (7 of 14) and F (3)”.*** (Lines 100-104)

“Most isolates (36 of 46) belonging to 9 of the 11 prevalent clones were carriers of ≥ 3 of specific traits chuA, fyuA, vat and yfcV associated with higher virulence, and therefore, positive for the UPEC status; and only clones D-ST349 (CH-54) and D-ST69 (CH35-27) were characteristically negative. Regarding the presence or absence of both iutA and kpsMII, we observed a positive correlation with isolates belonging to clones B2-ST73 (CH24-30), B2-ST131 (CH40-30), B2-ST404 (CH14-27), B2-ST1193 (CH14-64) and F-ST59 (CH32-41)”. (Lines 111-116).

Now, we have explained in the Discussion section, ***“In addition, we performed the screening of specific traits associated with the virulence status of isolates due to the strong correlation found previously between virulence-gene profiles and specific lineages of *E. coli* (Díaz-Jiménez et al. 2021; Flament-Simon et al., 2020a; Flament-Simon et al., 2020b).*** (Lines 239-242).

H) Line 301- The detailed inclusion of the specific clonal groups and numbers from a previous study, listing all of them is very detailed and atypical for the discussion. It would be better to just overview similarities and not delve into highly detailed comparisons in the discussion.

Much of the discussion restates what has been presented in the results. It would be of more interest to the readers to relate the findings of the study to those of other reports or describe what other studies have commonly identified that were also found in the current work. It would also be of value to include discussion concerning the discrepancies found between the UPEC/ExPEC gene screening which only indicated a likelihood of 2/3 of the strains to be either ExPEC or UPEC, and that there may be some UPEC clones that lack the sequences used in these simplified gene screens of only a few specific genes commonly linked mainly to B2 isolates.

Thank you for the comment. Following the suggestion, we have reduced the discussion and focused to overview similarities in this revised version (Lines 251-275). Also, we have clarified

the information regarding status (ExPEC and UPEC) throughout the text, as described in answers A to G.

i) Although accession numbers for genomes and sequence data are given in the text, the data has not been released for the accession numbers. Such sequence data should normally be available/accessible during the review process. In any case, this data should definitely be made immediately available prior to publication if the paper is accepted.

Thank you for the comment. Yes, the accession numbers are accessible, but the data will be release and available prior to publication, in the case this manuscript is accepted.

Minor corrections:

Line 75- replace "within this" with " to treat these", otherwise something is wrong or missing in this sentence. Thank you. Done (Line 86).

Line 250- should be "differentiate complicated UTI from uUTI. Thank you. Done (Line 184).

Line 252- should be giving and not given. Thank you. Done (Line 186).

Line 372- "importance" would be a more appropriate and accurate word than "necessity". Thank you. Done (Line 325).

Line 375- change to "pandemic ST1193 clonal group, which .." Thank you. Done (Line 330).

Reviewer #3 (Comments for the Author):

This study characterized 100 E. coli isolates from uncomplicated urinary tract infections from one health region in Spain. Various methods of strain typing were performed along with antimicrobial susceptibility testing. Five isolates belonging to ST1193 CH14-64 were subjected to whole genome sequencing and further analysis.

Major comments:

A) The study provided information to inform the local antibiogram of E. coli from uUTI in this particular health region.

This is useful information since the susceptibility pattern suggests that second line recommended therapies are more appropriate than the first line recommendation. This conclusion was consistent with several other larger studies that were cited.

Firstly, we want to thank the reviewer's comment highlighting the relevance of the study.

B) Meaningful conclusions cannot be drawn from the WGS portion of the manuscript as only 5 isolates were sequenced. Given that there were only 3 isolates in Enterobase matching this cgMLST (out of 1900 isolates of ST1198), this does not seem to be one of the prevalent strain types causing uUTIs in Spain or globally.

As far as we know, this is **the first report** on this new global emerging MDR clone B2-ST1193 (CH14-64) **in uncomplicated UTI**, and at least in Spain. Furthermore, **this is the most prevalent clone (prevalence 6%) of our study**, together with B2-ST141 (CH52-5). From our point of view, this finding is relevant to demonstrate that uUTI should be subject of surveillance due to the prevalence of such a clonal group associated with FQR and MDR (and typically positive for the status UPEC and ExPEC). **The first report in Spain was that of Flament-Simon et al., (2020a) with a prevalence of 1% in hospital samples.**

So, it is important to know the genomic characteristics of what seems a potentially spreading of resistance clone. Basically, **the genomic analysis of our typically lactose negative ST1193 isolates showed similar key genomic characteristics than those ST1193 clones disseminated worldwide.**

Presently (25th March), Enterobase displays 1,260 **ST1193** genomes. To investigate the phylogeny of our isolates in a global context, we used genomic data based on the cgMLST scheme and the hierarchical clustering scheme (HierCC) of Enterobase to retrieve three genomes belonging to the cgSTs 4085 and 72142. **HierCC, based on core genome multi-locus typing, allows incremental, static, multi-level cluster assignments of genomes. Most of the ST1193 genomes displayed in Enterobase are assigned into the same HierCC HC50 (26), which means all genomes in this cluster have links no more than 50 alleles apart. Furthermore, ESC_FA9684AA_AS (cgST4085), ESC_TA3850AA_AS and ESC_GB6748AA_AS (cgST72142) cluster together in HC20 (571) with no more than 20 alleles of difference.** The SNP-based phylogenetic tree comparison confirmed the high core genome similarity between the eight genomes (the three from Enterobase and five of the present study). **According to the metadata uploaded in Enterobase, this result would prove the fact of a conserved ST1193 clone circulating in different countries and years.**

In summary, we believe that there is a high conserved and prevalent clonal group ST1193 which is spread worldwide. Due to its recent emergence, it is relevant to trace its evolution through this kind of descriptive and comparative studies. In this new version, we have tried to explain better the genomic findings in the abstract (Lines 40-43).

C) Analysis of all isolates that were resistant to first line agents would be more informative. The discussion section was very thorough.

Thank you for the comment. Here, we aimed the characterization of the predominant uropathogenic *E. coli* (UPEC) lineages associated with uUTIs to assess the importance of implementing specific surveillance programs to treat these infections.

The detailed information of the whole collection is included in The Supplementary Table. The prevalent clones (≥ 3 isolates) and their characterization is included in the body of the manuscript.

Minor Comments

-Line 18-19. Specify that the 15 centres were from one health region.

Done (Line 27)

-Abstract: suggest highlighting high diversity of *E. coli* causing uUTI

Thank you for the suggestion. Done (Line 28)

-Line 80-81: if there was no potential selection bias, specify e.g. if you selected all samples in March and first 55 in June/July.

“To avoid bias, the first 100 isolates recovered during those periods of time (45 and 55, respectively) were included in the study.” This information has been added in the manuscript. (Lines 340-341)

-Line 100: briefly state what the method of Clermont is based on.

Done. The method has been briefly described in Lines 361-363.

-Line 110-111: I think amox/clav and pip/tazo are generally considered to part of the same drug class of penicillin + beta-lactamase inhibitor.

The reviewer is right, both amox/clav and pip/tazo are penicillins + beta-lactamase inhibitor. However, in the Magiorakos definition taken as reference in the present work, this group is divided into these two categories: antipseudomonal penicillins + beta-lactamase inhibitors and penicillins + beta-lactamase inhibitors.

-Line 166: Is it more correct to say that, "44% of E. coli were carriers of ONE OR both iutA and kpsMII"?

We meant that 44 isolates were carriers of both genes, iutA+kpsMII (please, see Supplementary Table). So, we have left it as it was in the original text.

REFERENCES

- Díaz-Jiménez D, García-Meniño I, Herrera A, Lestón L, Mora A. 2020. Microbiological risk assessment of Turkey and chicken meat for consumer: Significant differences regarding multidrug resistance, mcr or presence of hybrid aEPEC/ExPEC pathotypes of *E. coli*. *Food Control*, 107713.
- Flament-Simon S-C, Nicolas-Chanoine M-H, García V, Duprilot M, Mayer N, Alonso MP, García-Meniño I, Blanco JE, Blanco M, Blanco J. 2020a. Clonal Structure, Virulence Factor-encoding Genes and Antibiotic Resistance of *Escherichia coli*, Causing Urinary Tract Infections and Other Extraintestinal Infections in Humans in Spain and France during 2016. *Antibiotics*; 9(4):161.
- Flament-Simon S-C, García V, Duprilot M, Mayer N, Alonso MP, García-Meniño I, Blanco JE, Blanco M, Nicolas-Chanoine M.-H, Blanco J. 2020b. High Prevalence of ST131 Subclades C2-H30Rx and C1-M27 Among Extended-Spectrum β -Lactamase-Producing *Escherichia coli* Causing Human Extraintestinal Infections in Patients From Two Hospitals of Spain and France During 2015. *Front Cell and Infect Microbiol*, 10.
- Johnson JR, Murray AC, Gajewski A, Sullivan M, Snippes P, Kuskowski MA, Smith KE. 2003. Isolation and molecular characterization of nalidixic acid-resistant extraintestinal pathogenic *Escherichia coli* from retail chicken products. *Antimicrob Agents Chemother*, 47:2161-8.
- Magiorakos AP, Srinivasan A, Carey RB, Carmeli Y, Falagas ME, Giske CG, Harbarth S, Hindler JF, Kahlmeter G, Olsson-Liljequist B, Paterson DL, Rice LB, Stelling J, Struelens MJ, Vatopoulos A, Weber JT, Monnet DL. 2012. Multidrug-resistant, extensively drug-resistant and pandrug-resistant bacteria: an international expert proposal for interim standard definitions for acquired resistance. *Clin Microbiol Infect.*;18(3):268-281.
- Manges AR, Geum HM, Guo A, Edens TJ, Fibke CD, Pitout JDD. 2019. Global Extraintestinal Pathogenic *Escherichia coli* (ExPEC) Lineages. *Clin Microbiol Rev*, 32.
- Matsumura Y, Noguchi T, Tanaka M, Kanahashi T, Yamamoto M, Nagao M, Takakura S, Ichiyama S, on behalf of the 89th JAID BRG. 2017. Population structure of Japanese extraintestinal pathogenic *Escherichia coli* and its relationship with antimicrobial resistance. *J Antimicrob Chemother*, 72(4):1040-1049.
- Riley LW. 2014. Pandemic lineages of extraintestinal pathogenic *Escherichia coli*. *Clin Microbiol Infect*, 20:380-90.

- Riley LW. 2020. Extraintestinal Foodborne Pathogens. *Annu Rev Food Sci Technol*, 11:275-294.
- Spurbeck RR, Dinh PC, Jr., Walk ST, Stapleton AE, Hooton TM, Nolan LK, Kim KS, Johnson JR, Mobley HL. 2012. *Escherichia coli* isolates that carry *vat*, *fyuA*, *chuA*, and *yfcV* efficiently colonize the urinary tract. *Infect Immun* 80:4115-22.
- Tchesnokova V, Billig M, Chattopadhyay S, Linardopoulou E, Aprikian P, Roberts PL, Skrivankova V, Johnston B, Gileva A, Igusheva I, Toland A, Riddell K, Rogers P, Qin X, Butler-Wu S, Cookson BT, Fang FC, Kahl B, Price LB, Weissman SJ, Limaye A, Scholes D, Johnson JR, Sokurenko EV. 2013. Predictive diagnostics for *Escherichia coli* infections based on the clonal association of antimicrobial resistance and clinical outcome. *J Clin Microbiol*, 51:2991-9.

April 26, 2022

Dr. Azucena Mora
Universidade de Santiago de Compostela
Departamento de Microbioloxía e Parasitoloxía
Laboratorio de Referencia de E. coli (LREC)
Facultade de Veterinaria
Lugo 27002
Spain

Re: Spectrum00041-22R1 (Occurrence and genomic characterization of clone ST1193 clonotype (CH)14-64 in uncomplicated urinary tract infections caused by Escherichia coli in Spain)

Dear Dr. Azucena Mora:

Your manuscript has been accepted, and I am forwarding it to the ASM Journals Department for publication. You will be notified when your proofs are ready to be viewed.

Sincerely,

Sadjia Bekal
Editor, Microbiology Spectrum

Journals Department
Supplemental Material 4: Accept
Supplemental Material 3: Accept
Supplemental Material 2: Accept
Supplemental Material 1: Accept